# Intake of Vitamin B12 and Folate and Biomarkers of Nutrient Status of Women within Two Years Postpartum

**DOI:** 10.3390/nu14183869

**Published:** 2022-09-19

**Authors:** Yu Shen, Lichun Huang, Yan Zou, Danting Su, Mengjie He, Yueqiang Fang, Dong Zhao, Wei Wang, Ronghua Zhang

**Affiliations:** Department of Nutrition and Food Safety, Zhejiang Provincial Center for Disease Control and Prevention, Hangzhou 310051, China

**Keywords:** vitamin B12, folate, supplement, dietary diversity score, postpartum women

## Abstract

Background: Little is known about variation in vitamin B12 and folate status among Chinese women 2 years postpartum. This study assessed intake of vitamin B12 and folate and biomarkers of nutrient status among Chinese women postpartum. Methods: Demographic information, multi-/single-nutrient supplementation, dietary data, serum vitamin B12 and serum folate were assessed in 982 women within 2 years postpartum, using ten investigation sites in Zhejiang Province from the National Nutritional Study 2016–2017, which is a nationally representative cross-sectional study, to form a representative provincial sample of Zhejiang Province. The dietary diversity score (DDS) was used for assessing the dietary pattern. Results: Vitamin B12 increased slightly at the early stage of postpartum and then dropped over time. Serum folate level elevated with postpartum time. The median serum vitamin B12 concentration was 494.59 (373.21–650.20) pg/mL, and folate was 7.58 (5.02–10.34) ng/mL. Correspondingly, vitamin B12 levels suggesting marginal deficiency (200–300 pg/mL) and deficiency (<200 pg/mL) resulted as 9.27% and 3.26%, respectively, and folate level suggesting deficiency (<3 ng/mL) was 9.16%. Multi-/single-nutrient supplementation during pregnancy was associated with log-transformed serum vitamin B12 and folate level after adjusting for potential confounders (vitamin B12: ß (SE) = 0.124 (0.028), *p* < 0.001; folate: 0.128 (0.035), <0.001). Additionally, postpartum nutrient supplementation was associated with log-transformed serum folate level, especially for lactating women (ß (SE) = 0.204 (0.062), *p* = 0.001). Increased DDS was significantly associated with elevated serum vitamin B12 and folate levels (vitamin B12: ß (SE) = 0.028 (0.011), *p* = 0.011; folate: 0.030 (0.014), 0.031). In addition, age and educational level were influencing factors for serum vitamin B12 and folate concentrations among postpartum women. Conclusion: Serum vitamin B12 level decreased and folate level increased with postpartum age among Chinese women. Nutrient supplementation during pregnancy was related to elevated serum vitamin B12 and folate concentrations. Postpartum nutrient supplementation was associated with the increased serum folate level of lactating women. Dietary diversity was related to increased serum vitamin B12 and folate levels, especially among postpartum women with younger age and lower educational level.

## 1. Introduction

Vitamin B12 and folate deficiency/insufficiency in women is common in low- and middle-income countries [1]. Pre- and postpartum women are both at particularly high risk for vitamin B12 and folate insufficiency/deficiency due to the increased nutritional demands of the mother, the fetus and the infant during this period [2]. A Dutch study found that 9.9% of women within 5 weeks postpartum suffered from serum vitamin B12 deficiency (<180 pmol/L) [3]. Bellows et al. [4] revealed that 5.2% of mothers were suffering vitamin B12 (<203 pg/mL) deficiency and 25.6% with insufficiency (203–271 pg/mL) within 3 months postpartum in Tanzania. An American study found that 10.25% of mothers within 24–72 h postpartum suffered from plasma folate insufficiency/deficiency (<13.5 nmol/L) [5]. In China, a cross-sectional study in Guizhou and Henan provinces revealed that the prevalence of vitamin B12 (<200.0 pg/mL) and folate (<4.0 ng/mL) deficiency was 2.3% and 29.8% among lactating mothers [6].

Vitamin B12 and folate insufficiency has significant public health consequences, for both mothers and infants. Low vitamin B12 and folate status was found to be associated with anemia [7,8], abnormal nervous system [9] and glossitis [10]. Infants obtain vitamin B12 and folate from breast milk. Adequate vitamin B12 in infants is important for their growth and cognitive development, and its deficiency leads to failure to thrive, developmental regression and severe neuropathy [11]. Deficiency or impaired utilization of folate has been acknowledged as the cause of a number of complications, including megaloblastic anemia [12], cardiovascular disease [13] and neurological problems [14]. Additionally, folate is now known to play a much more profound and wide-ranging role in the development of neural tube defects (NTD) in neonates [15] and infants [16]. There is a positive correlation between maternal and infant blood folate concentration during lactation [17], and milk folate level is associated with maternal folate intake [18]. Therefore, more attention should be paid to vitamin B12 and folate insufficiency in women during lactation.

Dietary intake is the major source of vitamin B12 and folate in humans, through daily diet and nutrient supplementation. Typical dietary sources of vitamin B12 are animal products, for example, meat, egg, milk, fish and shellfish [19]. The main dietary sources of folate are green leafy vegetables, pulses, liver, egg, and fortified grain products [20]. Lactating mothers not consuming a high-quality diet could end up with a suboptimal intake of key nutrients essential to the health of themselves and their children. Women planning or able to get pregnant are recommended to take a folic acid supplement [21]; maternal prenatal micronutrient supplementation can often be seen in China nowadays [22]. However, little is known about nutrient supplement use for Chinese women postpartum. Hence, the purpose of this study was to assess biomarkers of vitamin B12 and folate status and to investigate the relationship between micronutrients, dietary diversity and the vitamin B12 and folate levels of women within 2 years postpartum in southeast China.

## 2. Materials and Methods

### 2.1. Study Design

This study was based on data obtained from the China National Nutrition and Health Survey 2016–2017 (CHNNS2016-2017). Our study chose women within 2 years postpartum from ten investigation sites in Zhejiang Province, including urban and rural areas, to form a representative provincial sample of Zhejiang Province to assess the nutritional status of postpartum women. In Zhejiang Province, the field investigation, physical examination and blood specimen collection were conducted between September 2016 and November 2017. The study was conducted in accordance with the Declaration of Helsinki and approved by the Institutional Review Board of the Chinese Center for Disease Control and Prevention (protocol code 201614 and approval date 3 June 2016). All women provided written informed consent after the research protocols were carefully explained to them. 

### 2.2. Sampling Method and Study Population

This survey is part of the Chinese National Nutrition and Health Survey [23]. The China National Nutrition and Health Survey in 2016–2017 was a cross-sectional survey designed to examine the health and nutritional status of children, adolescents and women that were 2 years postpartum. A multi-stage stratified cluster sampling design was used for the selection of participants. In Zhejiang Province, there were ten study sites representing urban and rural areas of provincial coverage. Then, two townships or subdistricts were randomly sampled from each study site. Among selected townships or subdistricts, two villages or communities were randomly sampled. Finally, women two years postpartum living in selected villages or communities were included and interviewed. At each survey site, at least 100 women were interviewed. In this study, the inclusion criteria of the women were: (1) agreed to participate in the study; (2) within 2 years after delivery. The exclusion criteria were: (1) genetic metabolic disease; (2) chronic cardiovascular and cerebrovascular diseases; (3) mental illness.

### 2.3. Data Collection

Women’s age, educational level, multi- and single-nutrient (including vitamin and mineral) supplementation during pregnancy and at present were collected by a general information questionnaire. Anthropometric measurements were conducted by trained health workers of the local community health center. BMI was calculated by weight (kg)/height (m)^2^ and then categorized into three groups (lean: <18.5; normal: 18.5–23.9; overweight/obesity: ≥24 kg/m^2^). Multi-nutrient supplementation referred to supplements that contained two or more micronutrients and single-nutrient supplementation referred to supplements that contained only one nutrient. Lactating women referred to those who were in lactation and breastfeeding, and non-lactating women included those who never breastfed or had breastfed but had already weaned their infant.

### 2.4. Dietary Diversity Measurement

The food frequency and intake questionnaire consisted of 11 items (staples, soybean products, vegetables, fruits, mushroom/algae, dairy products, meat, fish/seafood, egg, snacks, drinks) and oil/condiments, assessing the previous 7 days. According to the Dietary Guidelines for Chinese Residents [24] and the FAO DDS questionnaire [25], nine food groups were selected from all the food items, including cereal, soybean products, meat, egg, dairy products, fish/seafood, fat/oil, fruits and vegetables. A Dietary Diversity Score (DDS) point was added when at least one food item from the mentioned food groups was consumed. DDS was the aggregate score of all consumed food groups, ranging from 0 to 9 based on the structure of this guideline, and DDS was divided into two groups: low (≤6) and high (>6) [25].

### 2.5. Blood Sample Collection and Measurement of Serum Vitamin B12 and Folic Acid

Six milliliters of fasting venous blood were drawn. Serum B12 and folate levels were measured by a fully automated analyzer based upon the chemiluminescence immunometric assay method.

### 2.6. Definition of Vitamin B12 Insufficiency and Folate Deficiency

Serum vitamin B12 and folate concentrations were used to evaluate an individual’s biomarkers of vitamin B12 and folate status. Vitamin B12 deficiency was defined as serum level lower than 200 pg/mL, and marginal deficiency was defined as between 200 and 300 pg/mL according to the American Centers for Disease Control and Prevention (CDC) recommendation [26]. Due to the small number of vitamin B12-deficient women, deficiency and marginal deficiency were combined for insufficiency. Folate deficiency was defined as serum folate concentration less than 3 ng/mL according to the World Health Organization (WHO) recommendation [27].

### 2.7. Statistical Analyses

Mean and standard deviations (mean ± SD) were used to described continuous normal variables, and median and quartile (median (quartile)) were used for variables with skewed distribution. Frequency and percentage (%) were reported for the categorical variables. Continuous and categorical variables were compared using Student’s t-test and the chi-square test, respectively. Because of skewed distribution, serum vitamin B12 and folate concentrations were log-transformed. A multiple linear regression model and logistic model were used to detect the association between DDS, age, educational level, pregnancy BMI, and parity and serum vitamin B12/folate concentrations and vitamin B12 insufficiency/folate deficiency, respectively. These two models were also performed to investigate the relationship between multi-/single-nutrient supplement and vitamin B12/folate concentrations and insufficiency/deficiency. Additionally, the figures were plotted using postpartum month against the vitamin B12/folate concentrations. All statistical analyses were performed by using the program package R version 3.5.1. *p* less than 0.05 was considered statistically significant.

## 3. Results

A total of 982 women within 2 years postpartum were enrolled in the current study. The demographic characteristics of the participants are presented in Table 1. Overall, the median serum vitamin B12 concentration was 494.59 (373.21–650.20) pg/mL and folate was 7.58 (5.02–10.34) ng/mL. Vitamin B12 increased slightly at the early postpartum stage and then dropped over time. Serum folate level elevated by postpartum time and it increased rapidly in the first 10 months and then elevated moderately (see Figure 1). Correspondingly, 9.27% and 3.26% of women suffered from vitamin B12 concentration suggesting marginal deficiency and deficiency, and 9.16% suffered from folate concentration suggesting deficiency. As is shown in Table 2 older age (*p* = 0.039) and higher educational level (undergraduate or above: *p* = 0.014) were associated with increased log-transformed serum vitamin B12 concentration; older age (*p* < 0.001) and higher educational level (undergraduate and above: *p* = 0.007) were associated with increased log-transformed folate level, and higher BMI (≥24: *p* = 0.022) was related to a decreased level, respectively. Subjects with folate deficiency were older in age (*p* = 0.014) (see Appendix A
Table A1).

### 3.1. Nutrient Supplement and Serum Vitamin B12/Folate Status

During pregnancy, 51.73% of the participants took multi- or single-nutrient supplements. However, at the time of investigation, only 12.22% of the women took multi- or single-nutrient supplements. After adjustment for potential confounders, compared with those who did not take nutrient supplements during pregnancy, those with supplementation had higher log-transformed serum vitamin B12/folate concentrations (vitamin B12: ß (SE) = 0.124 (0.028), *p* < 0.001; folate: 0.128 (0.035), <0.001, respectively). Nutrient supplementation after delivery was associated with increased log-transformed serum vitamin folate level (ß (SE) = 0.141 (0.054), *p* = 0.009). However, they did not obtain the higher log-transformed serum vitamin B12 level (Table 3). Furthermore, among lactating women, postpartum nutrient supplementation was positively and significantly associated with folate level (ß (SE)= 0.204 (0.062), *p* = 0.001), and it had no effect on folate status among non-lactating women. Moreover, taking multi- or single-nutrient supplements during pregnancy reduced risks of vitamin B12 insufficiency and folate deficiency (vitamin B12: OR = 0.49, 95%CI: 0.33–0.73; folate: 0.57, 0.36–0.90) (Appendix A
Table A2).

### 3.2. Dietary Diversity and Serum Vitamin B12/Folate Status

The average DDS of women was 7.38 ± 1.62; in multiple linear regression analyses, DDS level was positively associated with log-transformed serum vitamin B12/folate concentration (vitamin B12: ß (SE) = 0.028 (0.011), *p* = 0.011; folate: 0.030 (0.014), 0.031). When DDS was divided into “low dietary status” (≤6) and “high dietary status” (>6), compared with low status, high status was positively correlated with log-transformed serum vitamin B12 concentration (ß (SE) = 0.077 (0.038), *p* = 0.042) (Table 4). For lactating women, the relationships between DDS and log-transformed serum vitamin B12 and folate levels were similar, but these relationships were not seen among non-lactating women. The multiple logistic regression demonstrated that increased DDS reduced the risk of folate deficiency (Appendix A
Table A2).

## 4. Discussion

Vitamin B12 and folate status have profound effects on complications related to lactation and infant growth. To date, few studies have described biomarkers of vitamin B12 and folate status during lactation or the postpartum period among Chinese women. This study investigated the vitamin B12 and folate levels of women 2 years postpartum and influencing factors, which are beneficial for assessing nutritional status in postpartum women.

The present study showed a moderate prevalence of serum vitamin B12 and folate deficiency in women within 2 years postpartum. Vitamin B12 increased slightly early on postpartum and then dropped over time. Serum vitamin B12 and folate concentrations in postpartum women have been reported in several studies previously. In a cross-sectional study from the Netherlands, the prevalence of vitamin B12 deficiency (<180 pmol/L) was 9.09% [3]. In a study from Tanzania, the prevalence of vitamin B12 (<203 pg/mL) deficiency was 5.2%, insufficiency (203–271 pg/mL) was 25.6% and the median concentration was 395.4 (286.8–523.5) ng/mL among women three months postpartum [4]. A Brazilian cohort study revealed that the prevalence of vitamin B12-deficient status (<148 pmol/L) was 3.9% and the median level was 310 (249–391) pmol/L at 28–50 days postpartum; at 88–119 days postpartum, the rate of deficiency was 0 and the median level was 283 (217–374) pmol/L [28]. In China, a cross-sectional study including 1976 women within 2 years postpartum showed that the median serum vitamin B12 level was 469.0 (349.0–633.5) pg/mL, the prevalence of vitamin B12 deficiency (<200 pg/mL) was 2.7% and the marginal deficiency (200–300 pg/mL) rate was 12.8% [29]. In another study from Guizhou and Henan including 309 lactating women, the level of vitamin B12 was 437.7 (418.7–457.6) pg/mL and 2.3% of them suffered deficiency (<200 pg/mL) [6]. Our results showed that serum vitamin B12 increased slightly in the first few months after delivery, reached its peak at the 10–15th month and then dropped rapidly with postpartum age. The median level of vitamin B12 was 494.59 (373.21–650.20) pg/mL among women within 2 years postpartum, and the rates of deficiency and marginal deficiency were 3.26% and 9.27%, respectively. The biomarkers suggested serum vitamin B12 status of postpartum women in Zhejiang province was sufficient.

The prevalence of folate deficiency differed worldwide. A survey from Brazil revealed that the rate of folate deficiency (<4 ng/mL) of lactating women was 9.9%, and the median level was 9.8 (7.6–12.2) ng/mL [30]. A study in America found that the prevalence of plasma folate insufficiency/deficiency (<13.5 nmol/L) among 24–72 h postpartum mothers was 10.25% [5]. However, another study from America found that mean red blood cell folate was highest among women in the first year postpartum, with overall mean levels of 606 ± 15 ng/mL, compared with those investigating other time periods of women’s reproductive lifespan [31]. In the study from Tanzania, the prevalence of folate deficiency was 19.0% and the average concentration was 6.1 (4.4–8.8) ng/mL among women three months postpartum [4]. The study from Guizhou and Henan showed that the level of serum folate was 5.39 (3.74–8.32) ng/mL and 29.8% of them suffered deficiency (<4 ng/mL) [6]. In our study, we found that serum folate level elevated with postpartum time. It increased rapidly in the first 10 months and then elevated moderately. The median level was 7.58 (5.02–10.34) ng/mL, higher than that from Guizhou and Henan, and the deficiency (<3 ng/mL) was 9.16%. More attention should be paid to folate status in Chinese postpartum women.

Micronutrient supplement use was common among pregnant and lactating women. The majority of pregnant women consumed supplements containing a wide range of nutrients, and some of them continued to take them after delivery. Women planning or able to get pregnant are recommended to take a supplement containing 400–800 μg of folic acid each day according to the U.S. Preventive Services Task Force [21]. Additionally, according to the US Food and Nutrition Board (FNB), there is no specific recommendation for vitamin B12 supplementation dose for pregnancy and lactation, and a range of 30 to 100 μg intake per day is provisionally recommended for adults [32]. A cross-sectional study in the US showed that 77.0% of pregnant and 70.3% of lactating women used one or more nutrient supplements, and 64.4% and 54.2% among them used multi-micronutrient supplements [33]. Our results showed that 51.73% of women took multi-/single-nutrient supplements during pregnancy. However, only 12.22% of them took multi-/single-nutrient supplements during the postpartum period. Moreover, our results for nutrient supplementation during pregnancy were significantly associated with increased serum vitamin B12 and folate levels at the postpartum stage, and reduced risks for vitamin B12 insufficiency and folate deficiency. However, nutrient supplementation during the postpartum period only had effects on serum folate concentration. After stratifying by lactation, the relationship only existed among lactating women. This might indicate that supplement use during pregnancy should be promoted, as it has a long-lasting effect until the postpartum stage. Additionally, for lactating women, nutrient supplementation is an effective way to improve folate status.

A dietary pattern contains more food sources of major nutrients. Healthy food patterns are composed of food groups rich in vitamin B12 and folate, and DDS is significantly correlated with micronutrient intake [34]. A longitudinal study from India showed that among young people aged 18 years, low DDS (≤4) was associated with plasma vitamin B12 deficiency (<150 pmol/L) (adjusted OR = 1.89, 95%CI 1.24–2.87), but was not related to plasma folate deficiency (<7 nmol/L) [35]. A cross-sectional study in Indonesia for assessing the nutrient intake status of lactating women 2–5 months postpartum showed that lactating mothers were at risk for inadequate micronutrient consumption, with a shortage of animal-source foods and fresh fruits and vegetables [36]. However, a cross-sectional study from Mozambique of adolescent girls did not find an association between DDS and serum folate level [37]. Our results showed that DDS was positively associated with serum vitamin B12 and folate concentrations, and high DDS status was related to increased serum vitamin B12 status. Additionally, a high DDS reduced risks for folate deficiency. This indicates that postpartum women should maintain a balanced diet to acquire sufficient nutrients.

Furthermore, in our results, older age and higher educational level were found to be associated with higher serum vitamin B12 and folate status after adjusting for BMI, parity, breastfeeding and postpartum age, but were not related to vitamin B12 insufficiency or folate deficiency. Some studies of adult women showed that risks for vitamin B12 and folate level increased with older age [38,39] and lower educational level [40,41], consistent with our results, but some did not draw this conclusion [42,43]. Breastfeeding has been a risk factor for lower vitamin B12 and folate levels among postpartum women in many studies [6,29,44,45], but this was not found in our study. This could indicate that women during lactation would pay more attention to nutrition intake and this offsets the nutrient consumption by milk.

There were several strengths in our study. Firstly, this study evaluated the serum vitamin B12 and folate biomarkers statuses of both lactating and non-lactating women, 2 years postpartum, while other studies have only focused on the vitamin B12 and folate levels of pregnant or lactating women. Secondly, the present study showed that nutrient supplementation during pregnancy and dietary diversity are important factors for vitamin B12 and folate levels. However, there were two limitations in this study. Firstly, due to the study design, we did not collect the milk of lactating women in order to assess vitamin B12 and folate status in more ways. Secondly, the nutrient supplement and dietary data were self-reported and might have been influenced by recall bias.

## 5. Conclusions

Serum vitamin B12 level decreased and folate level increased with postpartum age among Chinese women. Nutrient supplementation during pregnancy was related to elevated serum vitamin B12 and folate concentrations. Postpartum nutrient supplementation was associated with an increased serum folate level in lactating women. Dietary diversity was related to increased serum vitamin B12 and folate levels, especially among postpartum women of a younger age and lower educational level.

## Figures and Tables

**Figure 1 nutrients-14-03869-f001:**
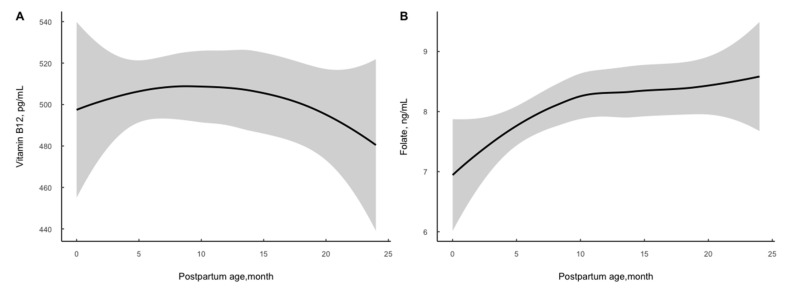
Distribution of serum vitamin B12 (**A**) and folate (**B**) concentration by postpartum age among 982 women.

**Table 1 nutrients-14-03869-t001:** Basic characteristics of women within 2 years postpartum.

Variables	Value
Participants (N)	982
Age of women, year *	30.96 ± 4.94
Postpartum age, month *	10.10 ± 6.48
Nationality ^#^	
Han	963 (98.07)
Others	19 (1.93)
Educational level ^#^	
Junior school or below	312 (31.77)
Senior high school	231 (23.52)
College	213 (21.69)
Undergraduate or above	226 (23.01)
BMI, kg/m^2^ *	22.68 ± 4.21
Parity ^#^	
1	462 (47.05)
≥2	500 (50.92)
Unknown	20 (2.04)
Alcohol ^#^	54 (5.50)
Smoking ^#^	4 (0.41)
Anemia during pregnancy ^#^	
Yes	335 (34.11)
No	605 (61.61)
Unknown	42 (4.28)
Breastfeeding and weaning ^#^	
Breastfeeding	550 (56.01)
Breastfed and weaning	370 (37.68)
Never breastfeed	62 (6.31)
Multi-/Single-nutrient supplement during pregnancy ^#^	
No	441 (44.91)
Yes	508 (51.73)
Unknown	33 (3.36)
Multi-/Single-nutrient supplement postpartum ^#^	
No	835 (85.03)
Yes	120 (12.22)
Unknown	27 (2.75)
Dietary Diversity Score (DDS) *	7.38 ± 1.62
DDS state ^#^	
Low (≤6)	182 (18.53)
High (>6)	800 (81.46)
Vitamin B12, pg/mL ^†^	494.59 (373.21–650.20)
Deficiency of vitamin B12 ^#^	
Marginal deficiency	91 (9.27)
Deficiency	32 (3.26)
Folic acid, ng/mL ^†^	7.58 (5.02–10.34)
Deficiency of folate ^#^	90 (9.16)

Vitamin B12: deficiency: <200 pg/mL; insufficiency: 200–300 pg/mL. Folate: deficiency: <3 ng/mL. * Values are mean ± SD; ^#^ values are N (%); ^†^ values are median (IQR).

**Table 2 nutrients-14-03869-t002:** The adjusted association of the relevant variables with vitamin B12 and folate concentrations by multiple regression model.

Variables	Vitamin B12	lg Vitamin B12		Folate	lg Folate	
	Median (IQR)	ß(SE)	*p*	Median (IQR)	ß(SE)	*p*
Age of women, year	494.59 (373.21–650.20)	0.007 (0.003)	0.039	7.58 (5.02–10.34)	0.023 (0.004)	<0.001
Educational level						
Junior school or below	487.21 (377.57–654.97)	ref	-	7.64 (4.8–10.23)	ref	-
Senior high school	468.61 (366.45–614.61)	−0.022 (0.038)	0.569	7.16 (4.98–10.34)	0.050 (0.048)	0.300
College	496.62 (360.49–627.60)	0.020 (0.040)	0.618	7.33 (5.11–9.24)	0.530 (0.050)	0.288
Undergraduate or above	558.73 (396.85–703.45)	0.099 (0.040)	0.014	8.04 (5.99–11.29)	0.136 (0.051)	0.007
BMI, kg/m^2^						
<18.5	515.43 (360.35–676.86)	ref	-	8.91 (5.55–11.11)	ref	-
18.5–23.9	500.81 (374.77–654.30)	−0.027 (0.053)	0.613	7.62 (5.11–10.44)	−0.085 (0.066)	0.199
≥24.0	491.34 (376.86–638.57)	−0.031 (0.056)	0.584	7.00 (4.83–9.55)	−0.163 (0.071)	0.022
Parity, times						
1	495.06 (354.57–650.40)	ref	-	7.45 (4.80–10.27)	ref	-
≥2	496.55 (392.90–649.66)	0.028 (0.033)	0.405	7.72 (5.35–10.36)	0.019 (0.042)	0.652
Unknown	436.27 (375.78–639.45)	0.035 (0.116)	0.764	6.96 (4.46–10.09)	0.067 (0.146)	0.644
Breastfeeding and weaning						
Breastfeeding	494.59 (377.98–648.68)	ref	-	7.49 (4.97–10.21)	ref	-
Breastfed and weaning	501.69 (362.86–661.64)	−0.008 (0.036)	0.832	7.81 (5.55–10.79)	−0.005 (0.046)	0.912
Never breastfeed	436.06 (376.35–633.19)	−0.038 (0.069)	0.589	6.96 (4.54–9.12)	−0.172 (0.087)	0.050

All above variables were included and adjusted for postpartum month.

**Table 3 nutrients-14-03869-t003:** The adjusted association of multi-/single-nutrient supplementation with vitamin B12 and folate levels by multiple linear regression model.

Multi-/Single-Nutrient Supplementation		Vitamin B12	lg Vitamin B12		Folate	lg Folate	
	*N*	Median (IQR)	ß (SE)	*p*	Median (IQR)	ß (SE)	*p*
During pregnancy							
All participants							
No/Unknown	474	457.24 (348.58–629.43)	ref	-	7.04 (4.48–10.08)	ref	-
Yes	508	526.80 (405.75–674.39)	0.124 (0.028)	<0.001	8.00 (5.58–10.71)	0.128 (0.035)	<0.001
Postpartum							
All participants							
No/Unknown	862	491.54 (370.26–641.71)	ref	-	7.40 (4.90–10.26)	ref	-
Yes	120	551.22 (395.03–711.97)	0.072 (0.043)	0.089	8.46 (6.36–1.082)	0.141 (0.054)	0.009
Lactating women *							
No/Unknown	459	492.02 (376.73–640.26)	ref	-	6.98 (4.77–9.92)	ref	-
Yes	91	536.00 (391.48–707.85)	0.049 (0.049)	0.315	8.85 (6.56–10.75)	0.204 (0.062)	0.001
Non-lactating women *							
No/Unknown	403	489.85 (362.45–644.93)	ref	-	7.75 (5.24–10.46)	ref	-
Yes	29	616.91 (418.67–741.27)	0.161 (0.085)	0.060	7.40 (6.08–10.79)	−0.039 (0.106)	0.717

Adjusted for: age, postpartum month, educational level, BMI, parity and breastfeeding or not. * Adjusted for: age, postpartum month, educational level, BMI and parity.

**Table 4 nutrients-14-03869-t004:** The adjusted association of dietary diversity with vitamin B12 and folate levels by multiple linear regression model.

		Vitamin B12	lg Vitamin B12		Folate	lg Folate	
	*N*	Median (IQR)	ß(SE)	*p*	Median (IQR)	ß(SE)	*p*
All participants							
DDS	982	494.59 (373.21–650.20)	0.028 (0.011)	0.011	7.58 (5.02–10.34)	0.030 (0.014)	0.031
Low (≤6)	182	458.52 (371.89–611.67)	ref	-	7.30 (4.52–10.42)	ref	-
High (>6)	800	520.51 (373.21–663.26)	0.077 (0.038)	0.042	7.62 (5.11–10.32)	0.038 (0.048)	0.430
Lactating women *							
DDS	550	494.59 (377.98–648.68)	0.034 (0.015)	0.025	7.50 (4.97–10.21)	0.050 (0.011)	0.011
Low (≤6)	83	457.78 (377.60–579.23)	ref	-	7.48 (5.01–10.24)	ref	-
High (>6)	467	502.71 (378.14–654.20)	0.088 (0.050)	0.082	7.53 (4.54–10.05)	0.032 (0.065)	0.624
Non-lactating women *							
DDS	432	494.32 (364.58–655.11)	0.023 (0.017)	0.182	7.75 (5.36–10.49)	0.014 (0.021)	0.516
Low (≤6)	99	459.27 (371.65–616.85)	ref	-	7.23 (4.50–10.50)	ref	-
High (>6)	333	502.30 (363.87–676.86)	0.070 (0.056)	0.217	7.78 (5.55–10.44)	0.057 (0.070)	0.417

Adjusted for: age, postpartum age, educational level, BMI, parity, breastfeeding or not. * Adjusted for: age, postpartum age, educational level, BMI, parity.

## Data Availability

The data presented in this study are available on request from the corresponding author. The data are not publicly available due to privacy and ethical concerns in this survey.

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
