# Peer review of "Intake of Vitamin B12 and Folate and Biomarkers of Nutrient Status of Women within Two Years Postpartum"

_nutrients, 2022, doi:10.3390/nu14183869_

Round 1

Reviewer 1 Report

I have included some comments to strengthen the manuscript that the authors might want to take into account while preparing a revision.

1 - In the sentence "A stratified multistage probability sampling design was used for the selection of participants" pages 90 and 91 I would like to see a reference about this sampling method used.  

2 - It's recommended to the authors do some tests to check residuals of multiple linear regression model. It's necessary the residuals satisfying some assumptions. The validity of the inference methods depends that. 

3 - It is not too much to recommend double checking the text to check possible errors and check the formatting in the tables (For example the first column of table 4 isn't clear to understand).

Reviewer 2 Report

This is a cross sectional study of Chinese women in Zhejiang Province based on a national study, extracting data for a single province. It evaluates the association among various factors on vitamin B12 and folate status in post-partum women. These factors include nutrient supplementation (multi-nutrient and single-nutrient during pregnancy and post-partum), dietary diversity score, and demographic factors such as age and education level.

The study methods are appropriate, and the background, results, and discussion are all presented appropriately. However, major editing is needed to get the English clear and meet typical conventions of speech. I started to provide more extensive comments on editing, but I think a copy editor can do this better than me.

There are a few key things: 1) the authors should say "nutrient supplementation" in place of "nutrient supplementary" throughout the manuscript. 2) In the methods, line 90, they mention that the Chinese National Nutrition and Health Study was designed to examine the health of "children and adolescents". This is a confusing statement since they evaluated Chinese women. I think this needs a little more explanation or revision.

Other small things:

line 10: say "this study assessed" in place of "is to assess".

line 27. Say "In addition", in place of "Besides"

Line 24. Delete second "was associated with".

Line 31. (and several places) make sure to say "was associated with" when describing relationships among serum folate or B12 and various factors. This is a cross sectional, so we can't say that any factor increased these levels, only that they were associated.

Line 58: delete random "f"

Line 100: "anthropometric" not "anthropometrical"

Line 193: say "few" in place of "little"

Everything else that needs to be fixed has to do mostly with verb tenses or use of prepositions.

Overall, the information in the study is useful and the authors put it in context well.
